# DocKS-RAG: Optimizing Document-Level Relation Extraction through LLM-Enhanced Hybrid Prompt Tuning

Xiaolong Xu [1]   Yibo Zhou [1]   Haolong Xiang [1]   Xiaoyong Li [2]   Xuyun Zhang [3]   Lianyong Qi [4]   Wanchun Dou [5]

## Abstract

Document-level relation extraction (RE) aims to extract comprehensive correlations between entities and relations from documents. Most of existing works conduct transfer learning on pretrained language models (PLMs), which allows for richer contextual representation to improve the performance. However, such PLMs-based methods suffer from incorporating structural knowledge, such as entity-entity interactions. Moreover, current works struggle to infer the implicit relations between entities across different sentences, which results in poor prediction. To deal with the above issues, we propose a novel and effective framework, named DocKS-RAG, which introduces extra structural knowledge and semantic information to further enhance the performance of document-level RE. Specifically, we construct a Document-level Knowledge Graph from the observable documentation data to better capture the structural information between entities and relations. Then, a Sentence-level Semantic Retrieval-Augmented Generation mechanism is designed to consider the similarity in different sentences by retrieving the relevant contextual semantic information. Furthermore, we present a hybrid-prompt tuning method on large language models (LLMs) for specific document-level RE tasks. Finally, extensive experiments conducted on two benchmark datasets demonstrate that our proposed framework enhances all the metrics compared with state-of-the-art methods.

[1]School of software, Nanjing University of Information Science and Technology, China [2]College of Meteorology and Oceanography, National University of Defense Technology, China [3]School of Computing, Macquarie University, Australia [4]College of Computer Science and Technology, China University of Petroleum (East China), China [5]State Key Laboratory for Novel Software Technology, Nanjing University, China. Correspondence to: Haolong Xiang <hlxiang@nuist.edu.cn>.

*Proceedings of the $42^{nd}$ International Conference on Machine Learning*, Vancouver, Canada. PMLR 267, 2025. Copyright 2025 by the author(s).

## 1. Introduction

Relation extraction (RE) is a critical task in natural language processing (NLP) domain, with substantial applications in real-world scenarios, such as knowledge graph construction, information retrieval, and automated question answering (He et al., 2022; Xiong et al., 2017; Nayak & Ng, 2019). The goal of RE is to extract structured and semantic triplets in the form of $subject - relation - object$ from the raw texts. With the rapid development of deep learning, substantial efforts have been dedicated to extracting the triplets within individual sentences, which is known as sentence-level RE. Although researchers have achieved remarkable advancements in sentence-level RE (Zeng et al., 2021; Lee et al., 2020; Baldini Soares et al., 2019), conversely, the domain of document-level RE remains under-explored.

Document-level RE poses greater challenges than sentence-level RE counterpart, primarily due to the presence of numerous implicit relations that require relational reasoning for identification. Moreover, entities may be related across texts, paragraphs, or even chapters within a document, rather than relying solely on extracting relations from isolated sentences, which requires comprehensive document-level understanding and precise extraction. For example, as illustrated in Figure 1, the subjects **"Wisconsin"** and **"the United States House of Representatives"**, along with the object **"US"**, appear in disparate sentences, complicating the identification of relations between them.

Existing document-level RE methods mainly conduct transfer learning on pre-trained language models (PLMs) (Huguet Cabot & Navigli, 2021; Xu et al., 2021b; Jiang et al., 2022; Zhou et al., 2020; Xu et al., 2025a), which focus on the representation of entities to better capture the contextual information within the documents. However, such PLMs-based methods suffer from modeling the interactions between entities and relations, which are known as the structural knowledge. For example, from the semantic meaning of the relation **"country"**, we can infer that the entities within the extracted triplet are related to locations. Ignoring such interactions hinders such PLMs-based methods from capturing the inner correlations between entities and relations, and leads to poor performance of extraction, especially in the complex document-level RE tasks.

*James William Murphy (April 17, 1858-July 11, 1927) was a U.S. Representative. (...2 sentences ...) He was elected mayor of Platteville for a two-year term in 1904, and was then elected to the United States House of Representatives as a Democrat in 1906, defeating Joseph W.Babcock for the seat from Wisconsin's 3rd congressional district.*

**Subject:** *United States House of Representatives*
**Object:** *U.S.*
**Relation:** country; applies to jurisdiction

**Subject:** *Wisconsin*
**Object:** *U.S.*
**Relation:** country; located in the administrative territorial entity

*Figure 1.* An example of cross-text relation extraction from raw documents.

To incorporate the rich structural knowledge for better extraction, several scholars consider constructing extra graphs to model the structural interactions within the documents, and then perform inference with the constructed graphs (Li et al., 2020; Nan et al., 2020; Qin et al., 2021; Zeng et al., 2020). Despite such graph-based methods pay more attention to the structural information, several limitations persist. Firstly, most of these methods lack of sufficient contextual information, which limits the semantic understanding of models. Moreover, most of such graph-based methods introduce extra noises and irrelevant connections, which will bring erroneous inferences and restrict the performance of extraction. Next, several works have been proposed to integrate pre-trained language models (PLMs) with additional knowledge graphs (KGs), aiming to enhance the models' global understanding across both structural and semantic spaces more effectively(Tang et al., 2020b; Fu et al., 2019; Sahu et al., 2019; Schlichtkrull et al., 2018; Xu et al., 2025b). However, PLMs are mostly trained on massive unstructured and contextual documentation data, while KGs are typically composed of structured and relational data. Such discrimination leads to the semantic misalignment between texts and graphs, which hinders PLMs from representing and utilizing structural knowledge within KGs, especially in complicated document-level RE tasks.

To address the issues outlined above, we propose a novel and effective framework, named DocKS-RAG, which aims to improve large language models (LLMs), such as LLAMA3 and GPT-4 (Dubey et al., 2024; Achiam et al., 2023) to perform document-level relation extraction (RE) tasks by introducing additional structural knowledge and enriched semantic information. Specifically, to better capture the rich interactions between entities and relations, we construct a Document-level Knowledge Graph (DocKG) from the observable documentation data. Then, a Sentence-level Se-

mantic Retrieval-Augmented Generation (SetRAG) mechanism is designed to consider the similarity of different sentences by retrieving the relevant contextual semantic information. From the perspective of semantic alignment, we transfer the structural knowledge from DocKG to informative expressions, and generate the hybrid prompts with the retrieved semantic information from SetRAG. Furthermore, we present a hybrid-prompt tuning method, which conducts Parameter-Efficient Fine-Tuning (PEFT) (Hu et al., 2021) to enhance the adaptability and performance of LLMs on specific document-level RE tasks. Finally, extensive experiments are conducted on two open benchmark datasets, DocRED (Yao et al., 2019) and Re-DocRED (Tan et al., 2022a), to further evaluate the performance of our proposed framework.

The main contributions of this paper are summarized as follows:

- We investigate the existing challenges in document-level RE task and propose a novel framework DocKS-RAG, which improves the performance of LLMs in document-level RE by considering the extra structural and semantic information.

- We design a sentence-level semantic Retrieval-Augmented Generation mechanism aimed at enhancing the semantic understanding of LLMs by retrieving the relevant sentences within the documents.

- We design a hybrid prompts generation method to combine the relevant structural interactions with semantic information from the constructed document-level knowledge graph and sentence-level knowledge base.

- We conduct extensive experiments on two benchmark datasets. The experimental results show that our proposed framework consistently outperforms the competitive methods.

## 2. Related Work

### 2.1. PLMs-based Methods for Document-Level Relation Extraction

Traditional pre-trained language models (PLMs), such as BERT (Devlin et al., 2019), have been widely applied in relation extraction (RE). REBEL (Huguet Cabot & Navigli, 2021) shows how RE can be simplified by expressing triplets as a sequence of text, and performs end-to-end RE for different relation types. Eider (Xie et al., 2022) transfers the attention from the entire documents to the subset of relevant sentences to improve both computational efficiency and extraction performance. RE-Flex (Goswami et al., 2020) tackles the challenge of unsupervised relation extraction by employing contextual matching. Later works achieve

advancement by integrating the structural knowledge to improve the global understanding of such PLMs-based models. A-GCN (Qin et al., 2021) proposes attentive graph convolutional networks to improve the performance of RE by building the context graph. GAIN (Zeng et al., 2020) constructs two extra graphs to capture the interactions among different mentions for accurate extraction. However, most of existing PLMs-based methods mainly focus on entity representations, while ignoring the rich interactions between entities and relations. Moreover, despite later works focus on introducing structural knowledge as priors, these methods still suffer from the semantic misalignment between PLMs and graphs, which leads to poor performance on document-level RE tasks.

### 2.2. Large Language Models for Document-Level Relation Extraction

With the rapid development of deep learning, large language models (LLMs) have revolutionized the natural language processing (NLP) domain (Chia et al., 2022; Soman et al., 2023; Arsenyan et al., 2024). Recently, researchers have dedicated to utilizing LLMs for document-level RE tasks. AutoRE (Xue et al., 2024) introduces a novel RE paradigm on LLMs, which eliminates the reliance on pre-defined relations and focuses on the triplet facts that are distributed across documents. GAP (Chen et al., 2024) proposes a generative context-aware prompt-tuning method to promote the adaptability of LLMs. GenRDK (Sun et al., 2024) generates extra auto-labeled data and employs a denoising strategy for LLMs to enhance the extraction performance. KG-RAG (Soman et al., 2023) addresses the challenges of domain adaptation by optimizing the context extraction process, which allows LLMs to generate the accurate texts from the established knowledge. However, such newly proposed methods primarily leverage the powerful semantic understanding capabilities of LLMs, while still largely ignoring the inherent heterogeneity between graph knowledge and the language models.

In general, previous PLM-based methods combat the effective incorporation of interactions between entities and relations for better extraction. Currently, recent proposed LLM-based works still fail to alleviate the misalignment between structural knowledge and semantic information.

## 3. Problem Definition

The primary objective for the general document-level RE task is to identify and classify the relations among entities mentioned throughout an entire document. Specifically, it aims to use annotated data from a set of observable documents $D_u$ to train a model $C$ that can accurately extract relations between entities in unobservable documents $D_s$, where $D_u \cap D_s = \emptyset$.

Formally, Given a document $D$ comprising a sequence of sentences $S = \{s_1, s_2, \ldots, s_n\}$, where $s_i$ represents the $i$-th sentence in the document, and the relation set $R = \{r_1, r_2, \ldots, r_M\}$ is pre-defined. We aim to identify all the entities $E = \{e_1, e_2, \ldots, e_k\}$, and extract all the possible triplets $T$ from $S$. Each triplet $T_j$ can be represented as $[e_a, r_m, e_b]$, where $e_a, e_b \in E$ are entities within the document, and $r_m$ denotes the category of relation between these entities. The objective function can be expressed as:

$$\mathcal{L} = - \sum_{(e_a, e_b) \in \mathcal{P}} \sum_{m=1}^{M} [y_{abm} \log(P(e_a, e_b | r_m))], \quad (1)$$

where $\mathcal{P}$ represents the set of all unique pairs of entities $(e_a, e_b)$ identified from the document. $y_{abm}$ is a binary indicator variable, which is set to 1 if the relation $r_m$ exists between entities $e_a$ and $e_b$, and 0 otherwise. $P(e_a, e_b | r_m)$ is the predicted probability that the relation $r_m$ exists between the entities $e_a$ and $e_b$. We aim to minimize the loss $\mathcal{L}$ between the predicted probabilities and the corresponding true labels to further identify relation among entity pairs.

## 4. Methodology

### 4.1. Overview

In this section, we will introduce the details of our proposed DocKS-RAG framework, which is as shown in Figure 2. Firstly, we construct the Document-level Knowledge Graph (DocKG) with Graph Neural Networks (GNNs) to obtain both representations of entities and relations. Secondly, a Sentence-level Semantic Retrieval-Augmented Generation (SetRAG) mechanism is designed to enrich the model's contextual understanding gained from the relevant documents. Finally, we present a hybrid-prompt tuning method, which generates the combined prompts from DocKG and SetRAG, and integrates by conducting Parameter-Efficient Fine-Tuning (PEFT) to further improve the performance of LLMs on document-level RE tasks.

### 4.2. Document-Level Knowledge Graph Construction for Hybrid Prompts Generation

In this part, we construct a document-level knowledge graph (DocKG) from the observable documentation data to obtain both entity and relation representations to further capture the topological information between them, which are presented in line 3 in Algorithm 1. Specifically, we employ GNNs, which are dominate approaches for constructing knowledge graphs (Li et al., 2020; Qin et al., 2021), to model the structural interactions between relations set $R$ and entities set $E$ within the documents $D$.

**Entity and Relation Representations.** To overcome the heterogeneity with language model augmented entity representations, we focus on modeling the interactions between

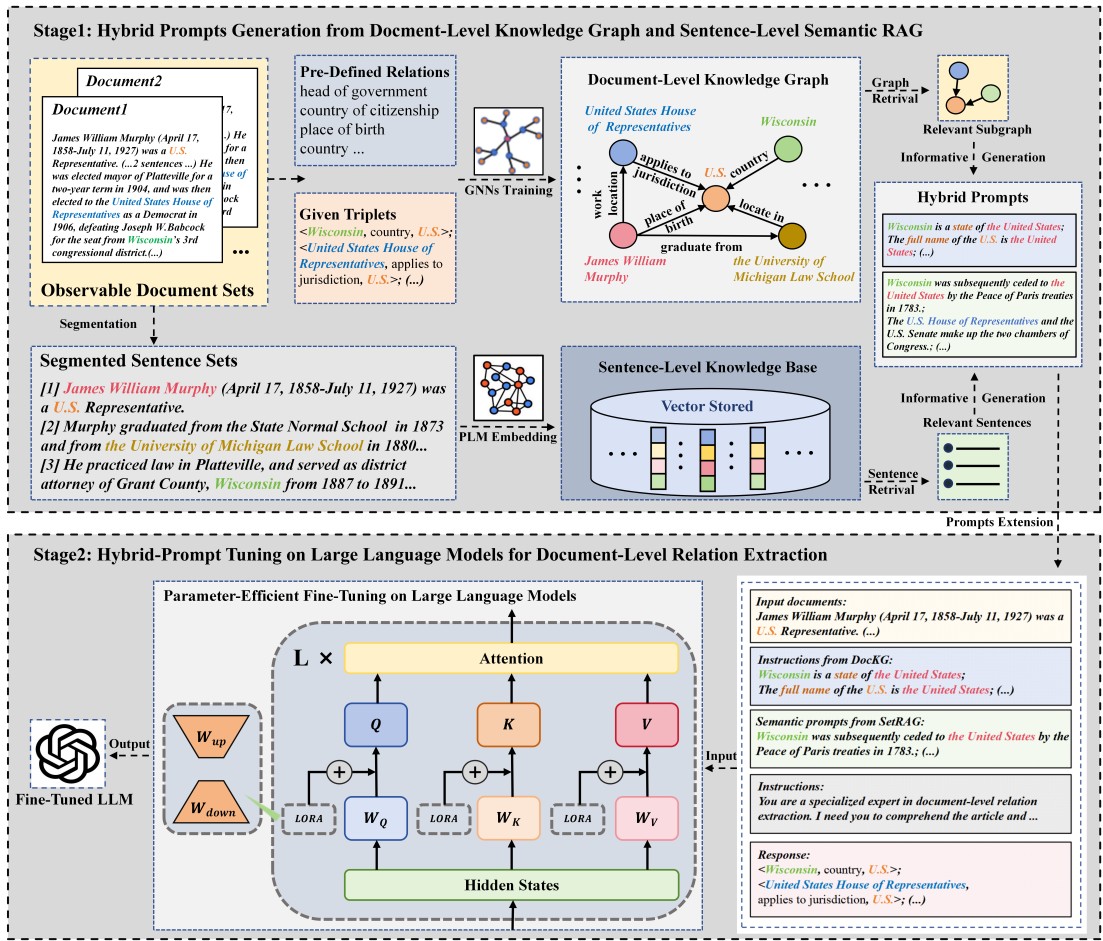

*Figure 2.* Overview of our proposed DocKG-RAG framework. It contains two key stages: (1) Hybrid prompts generation from both document-level knowledge graph (DocKG) and sentence-level semantic RAG (SetRAG); (2) Hybrid-prompt tuning based on large language models (LLMs) with LORA.

entities and relations to better capture the contextual information within the documents. Formally, in a general graph $G(E, R)$, each entity $e \in E$ is represented as an embedding vector $\mathbf{h_e^{(0)}} \in \mathbb{R}^d$, where $d$ is the dimension of the embedding size. Similarly, relations are represented by vectors that are embedded as $\mathbf{h_r^{(0)}} \in \mathbb{R}^d$. Then, we employ GNNs for document-level graph modeling and representation learning. In the training stage, entity and relation representations are iteratively updated by aggregating the neighbours' information. The update rule at layer $l$ is represented as:

$$\mathbf{h_e^{(l)}} = \sigma \left( \sum_{e' \in \mathcal{N}(e)} \mathbf{W_e} \mathbf{h_{e'}^{(l-1)}} + \mathbf{b} \right), \qquad (2)$$

$$\mathbf{h_r^{(l)}} = \sigma \left( \mathbf{W_r} \mathbf{h_r^{(l-1)}} + \sum_{(u,v) \in E} \mathrm{agg}(\mathbf{h_u^{(l-1)}}, \mathbf{h_v^{(l-1)}}) \right), \qquad (3)$$

where $\mathcal{N}(e)$ represents the set of neighboring entities of $e$, $\mathbf{W_e}$ is a learnable weight matrix of the entity embedding, $\mathbf{b}$ denotes a learnable bias vector, and $\sigma$ is referred to a non-linear activation function. $\mathbf{W_r}$ is the learnable weight matrix for the relationship embedding, and the summation is over all pairs of entities $u$ and $v$ with a relationship in the set $E$. The agg function aggregates the embeddings $\mathbf{h_u}$ and $\mathbf{h_v}$ of the entities at layer $l - 1$.

**Document-Level Knowledge Graph Construction and Relation Prediction.** After $L$ rounds of training with GNNs, the representations are utilized for relation prediction between the entities. For a pair of entities $(e_a, e_b)$, we compute their interaction scores as follows:

$$I(e_a, e_b) = \mathbf{h_{e_a}^{(L)}} \cdot \mathbf{h_{e_b}^{(L)}} + \mathbf{h_r^{(L)}}, \qquad (4)$$

where dot-product is conducted between entity representation $\mathbf{h_{e_a}^{(L)}}$ and $\mathbf{h_{e_b}^{(L)}}$, and $\mathbf{h_r^{(L)}}$ is the representation for the

**Algorithm 1** Hybrid-Prompt Tuning on Large Language Models

1: **Input:** Observable documents $D_u$, Given triplets $T_u$, Graph model $G_m$, Embedding model $E_m$, Informative generation function $Gen$, Recycling times $T$, Parameters of adapter $A$, User query $Q$.
2: **Output:** Predicted triplets $T_p$.
3: $KG \leftarrow Train(G_m, T_u)$
4: Segment $D_u$ into individual sentences and obtain the sentence sets $S_u$.
5: $KB \leftarrow E_m(S_u)$
6: Retrieve the relevant subgraph $G_q$ and TopK sentences $S_k$ from $KG$ and $KB$ according $Q$.
7: $P_g \leftarrow Gen(G_q)$, $P_s \leftarrow Gen(S_k)$
8: $P_h \leftarrow Concat(P_g, P_s)$
9: **for** $t = 1$ to $T$ **do**
10:     $LLM_{new} \leftarrow Train(LLM_{old}, P_h, A)$
11:     Update parameters of adapter $A$
12: **end for**
13: $T_p \leftarrow LLM_{new}(\hat{P}_h)$
14: **Return:** $T_p$

relation between $e_a$ and $e_b$. Then, we employ a softmax function to calculate the predicted probability of the relation type $r_j$ between $e_a$ and $e_b$, the process of calculation is formulated as:

$$P(r_j|e_a, e_b) = \frac{\exp(I(e_a, e_b, r_j))}{\sum_{r_k \in R} \exp(I(e_a, e_b, r_k))}. \qquad (5)$$

Finally, we construct the document-level knowledge graph, the construction of $DocKG$ is processed as:

$$DocKG = \{(e_a, r_j, e_b)|P(r_j|e_a, e_b) > \tau_{er}\}, \qquad (6)$$

where $\tau_{er}$ is a pre-defined threshold to determine the minimal probability of relation $r_j$ between entities $e_a$ and $e_b$.

In this way, we effectively leverage GNNs to capture the rich interactions between entities and relations within different documents. Meanwhile, the constructed DocKG consists of comprehensive representations in the semantic space, which enables more accurate extraction for document-level RE.

**Graph Retrieval for Hybrid Prompts Generation.** To efficiently extract structural knowledge from DocKG, we employ a graph retrieval method for further hybrid prompts generation. The objective of graph retrieval is to identify a relevant subgraph $G_q \subseteq DocKG$ associated with the given query $q$, and then obtain the informative prompts $P_{DocKG}$ based on the retrieved subgraph.

Specifically, we firstly transform the query $q$ into the semantic vector representation $\mathbf{h_q}$ by a pre-trained language model (PLM). Due to the differences in the embedding

spaces between PLM and DocKG, in our proposed method, we integrate PLM embeddings as initial inputs for entities and relations, allowing DocKG to learn information from the PLM through the process of updating and improve the reliability of similarity calculations. The semantic transformation is represented as:

$$\mathbf{h_q} = PLM(q). \qquad (7)$$

Then, for each entity $e \in E$, we further compute the similarity between $\mathbf{h_e}$ and $\mathbf{h_q}$ with a similarity function $\text{sim}(\cdot, \cdot)$, which is calculated by cosine similarity as follows:

$$\text{sim}(q, e) = \frac{\mathbf{h_q} \cdot \mathbf{h_e}}{\|\mathbf{h_q}\| \|\mathbf{h_e}\|}. \qquad (8)$$

Based on the computed similarities, we select the most relevant entities from DocKG and construct the candidate subgraph $G_q$ as:

$$G_q = \{e \in V \mid \text{sim}(q, e) > \tau_{eq}\}, \qquad (9)$$

where $\tau_{eq}$ is a pre-defined threshold that determines the retained sets of entity and relation interactions that are most relevant with the query $q$. Then, we create the informative prompts using the structural interactions within the retrieved subgraphs, along with a generation function $g_{DocKG}$, and the process of the informative prompts $P_{DocKG}$ generation is represented as:

$$P_{DocKG} = g_{DocKG}\Big(\{e, r|e, r \in G_q\}\Big). \qquad (10)$$

By adopting the proposed DocKG retrieval method, we effectively exploit the structural interactions between different entities and relations. Moreover, the relevant subgraphs associated with the user query are efficiently extracted and transferred into informative prompts, which further mitigates the semantic misalignment between graphs and texts.

### 4.3. Sentence-Level Semantic RAG Enhancement

In this part, to further enhance the semantic understanding of LLMs for document-level RE, we design a sentence-level semantic Retrieval-Augmented Generation (SetRAG) mechanism, which are presented in lines 4 to 5 in Algorithm 1. It mainly involves three processes for documents: segmentation, embedding and matching. Specifically, individual sentences are firstly segmented from the raw texts. Then, we construct the sentence-level knowledge base (SetKB) by embedding such sentences to the vector representations. Finally, relevant contextual sentences in the given document $D$ will be retrieved by similarity measures and incorporated into the hybrid prompts.

**Sentence Segmentation.** For a given document $D$, we aim to segment it into individual sentences. The process of segmentation is formulated as:

$$S = \text{Segment}(D) = \{s_1, s_2, \dots, s_n\}, \qquad (11)$$

where $S$ is the set of segmented sentences, and $n$ is the total number of extracted sentences. The segmentation function Segment utilizes punctuation (e.g., periods, exclamation marks) and linguistic rules to identify sentence boundaries.

**Sentence Embeddings and Knowledge Base Construction.** After we obtain the segmented sentences set, we further propose a sentence-level knowledge base (SetKB), which is constructed as a collection of embeddings that represent the semantic meaning of sentences. Specifically, each sentence $s_i \in S$ is embedded into a vector representation by a pre-trained language model (PLM), the construction of $SetKB$ is processed as:

$$\mathbf{t_i} = PLM(s_i), SetKB = \{(s_i, \mathbf{t_i}) \mid i \in n\}, \quad (12)$$

where each sentence $s_i$ is converted into its semantic embedding vector $\mathbf{t_i}$ and stored in the constructed $SetKB$.

**Knowledge Base Retrieval Process.** The constructed SetKB enables semantic retrieval of relevant sentences associated with the user queries. Specifically, given a query $q$, the most relevant sentences are identified and retrieved through similarity measuring. The calculation of measuring is formulated as:

$$M(q, \mathbf{t_i}) = sim\Big(PLM(q), \mathbf{t_i}\Big), t_i \in SetKB, \quad (13)$$

where $sim(\cdot, \cdot)$ is a similarity function. The *topk* relevant sentences from the SetKB are then retrieved. The process of the retrieval is expressed as:

$$S_k = Topk\Big(M(q, \mathbf{t_i}), k\Big). \quad (14)$$

Finally, we further generate another part of hybrid prompts $P_{SetRAG}$ using the retrieved sentences from SetKB, $S_k$, and theirs relevant triplets $T_k$. The generation of $P_{SetRAG}$ is represented as:

$$P_{SetRAG} = g_{SetRAG}\Big(\{T_i | T_i \in T_k\}\Big) + S_k. \quad (15)$$

The integration of sentence-level semantic RAG enhances the LLM's ability to accurately extract relations by providing rich and relevant semantic information, which empowers the semantic understanding of LLMs and ultimately improves the performance of document-level RE.

### 4.4. LLM-Enhanced Document-Level Relation Extraction through Hybrid Prompt Tuning

In this part, we will give the detailed description of our designed hybrid-prompt tuning framework that leverages both document-level Knowledge Graph and sentence-level semantic Retrieval-Augmented Generation (RAG) and conducts Parameter-Efficient Fine-Tuning (PEFT) to enhance the performance of LLMs on document-level RE tasks, which are presented in lines 6 to 12 in Algorithm 1.

**Hybrid Prompts Generation.** For a query $q$, we generate the hybrid prompts $P_{Hybrid}$, incorporating both relevant structural and semantic information from DocKG and SetRAG. The construction of hybrid prompts is expressed as:

$$P_{Hybrid} = Concat\Big(P_{DocKG}(q), P_{SetRAG}(q)\Big), \quad (16)$$

where $P_{DocKG}(q)$ represents the relevant structural knowledge retrieved from DocKG, and $P_{SetRAG}(q)$ represents the *Topk* relevant semantic information retrieved from SetRAG. The specific calculations are referred to Equs.(7-10) and Equs.(13-15).

**Parameter-Efficient Fine-Tuning on LLMs.** As we konw, LLM is initially fine-tuned on a large, annotated dataset and consists of massive generalized knowledge (Goswami et al., 2020). Subsequently, In our proposed DocKS-RAG framework, we adopt $LORA$, a general adapter training method (Hu et al., 2021), to fine tune LLMs. Specifically, by integrating adapter modules, we are supposed to achieve fine-tuning without modifying the entire parameter set of the pre-trained LLM. Moreover, the adapter modules $A$ are trained on smaller, task-specific datasets on document-level RE, which allows for rapid adaptation and minimizes the computational complexity of our proposed framework. The output with PEFT is calculated as:

$$O_{PEFT} = LLM\Big(P_{Hybrid}; \theta + \mathbf{A}\Big), \quad (17)$$

where $\theta$ are the original parameters of the pre-trained LLM, and $\mathbf{A}$ is referred to the learnable parameters associated with the adapter module that will be fine-tuned.

In summary, we propose a novel and effective framework, named DocKS-RAG, which mainly consists of two stages. In stage 1, we aim to generate the informative hybrid prompts by combining the relevant retrieved document-level structural knowledge and rich semantic information, which are based on the constructed document-level knowledge graph and sentence-level knowledge base. In stage 2, we consider conducting parameter-efficient fine-tuning on LLMs with the extended hybrid prompts to further enhance the adaptability and performance of LLMs on the complex document-level RE tasks.

## 5. Experiments

### 5.1. Datasets.

We evaluate the performance of our proposed DocKS-RAG on two widely used benchmark datasets: DocRED (Yao et al., 2019) and Re-DocRED (Tan et al., 2022a). We refer the reader to Appendix A.1 for more detailed description of both the datasets.

*Table 1.* Comparison of performance on DocRED and Re-DocRED datasets, showing both development (Dev) and test (Test) results with Ign-F1 and F1 scores. The best results are in bold and the second best is underlined "-". Our proposed DocKS-RAG outperforms other methods on both datasets.

| Model | DocRED | | | | Re-DocRED | | | |
|---|---|---|---|---|---|---|---|---|
| | Dev | | Test | | Dev | | Test | |
| | Ign-F1 | F1 | Ign-F1 | F1 | Ign-F1 | F1 | Ign-F1 | F1 |
| GEDA (2020) | 54.52 | 56.16 | 53.71 | 55.74 | - | - | - | - |
| LSR (2020) | 52.43 | 59.00 | 56.97 | 59.05 | - | - | - | - |
| GAIN (2020) | 59.14 | 61.22 | 59.00 | 61.24 | 71.99 | 73.49 | 71.88 | 73.44 |
| HIN (2020) | 54.29 | 56.31 | 53.70 | 56.60 | - | - | - | - |
| HeterGSAN (2021) | 58.13 | 60.18 | 57.12 | 59.45 | - | - | - | - |
| ATLOP-BERT (2021) | 59.22 | 61.09 | 59.31 | 61.30 | 73.35 | 74.22 | 73.23 | 74.02 |
| SSAN (2021) | 56.68 | 58.95 | 56.06 | 58.41 | - | - | - | - |
| SIRE (2021) | 59.82 | 61.60 | 60.18 | 62.05 | - | - | - | - |
| DocuNet (2021) | 59.86 | 61.83 | 59.93 | 61.86 | 73.60 | 74.62 | 73.53 | 74.48 |
| KMGRE (2022) | - | - | - | - | 73.33 | 74.24 | 73.39 | 74.46 |
| KD-DocRE (2022) | 60.08 | 62.03 | 60.04 | 62.08 | 73.68 | 74.66 | 73.64 | 74.55 |
| SRLR (2023) | 60.02 | 62.14 | 59.74 | 61.88 | - | - | - | - |
| AutoRE (2024) | - | - | - | - | - | 54.29 | - | 53.84 |
| LMRC (2024) | 59.22 | 61.08 | 58.62 | 60.69 | - | - | 72.35 | 72.93 |
| **DocKS-RAG(Ours)** | **62.76** | **63.87** | **61.83** | **63.59** | **74.64** | **75.31** | **74.32** | **75.49** |

## 5.2. Baselines.

We evaluate our proposed framework by comparing with recently presented PLMs-based, graph-based and LLMs-based methods, including HIN (Tang et al., 2020a), ATLOP-BERT (Zhou et al., 2020), SRLR (Huang et al., 2023), GLRE (Wang et al., 2020), LSR (Nan et al., 2020), HeterGSAN (Xu et al., 2020), SIRE (Zeng et al., 2021), SSAN (Xu et al., 2021a), DocuNet (Zhang et al., 2021), KD-DocRE (Tan et al., 2022b), KMGRE (Jiang et al., 2022), AutoRE (Xue et al., 2024), and LMRC (Li et al., 2024). We refer the reader to Appendix A.2 for more details of the baselines.

## 5.3. Implementation Details.

The technical modules of DocKS-RAG primarily consist of document-Level knowledge graph construction, sentence-level knowledge base construction, and hybrid-prompt tuning. The parameters for the comparative experiments are set as reported in their paper. We refer the reader to Appendix A.3 for more details of the experimental parameter settings.

## 5.4. Metrics.

In general document-level RE tasks, both the F1 and Ign-F1 scores play crucial roles in evaluating document-level RE methods. The F1 score allows for a balanced assessment, where both false positives and false negatives are taken into account. On the other hand, the Ign-F1 score (Ignored F1 score) provides a clearer picture of a model's effectiveness in extracting the implicit triplets from the documents. In this paper, we employ both F1 and Ign-F1 scores as our evaluation metrics, where higher values indicate better performance in document-level RE.

## 5.5. Results Analysis

**Document-Level Relation Extraction.** Table 1 summarizes the performance of DocKS-RAG comparing with various methods on DocRED and Re-DocRED datasets, and the advanced conclusions are summarized as follows:

Firstly, comparing with previous PLMs-based methods, we conduct graph retrieval on an extra constructed document-level Knowledge Graph (DocKG) to better capture the structural knowledge within the documents. We later generate the informative prompts from the retrieved subgraphs, which mitigates the heterogeneity between structural graph knowledge and the language models. Secondly, comparing with recent proposed graph-based methods, we consider the similarity between sentences, and further design a sentence-level semantic Retrieval-Augmented Generation (SetRAG) mechanism to enrich the contextual understanding of models. Furthermore, we present a hybrid-prompt tuning method, which generates the hybrid informative prompts by retrieving the relevant information both from DocKG and SetRAG, integrating with an adapter training method further enhances the adaptability of LLMs for better extraction. Finally, the experimental results demonstrate that our proposed DocKS-RAG outperforms competitive methods in both Ign-F1 and F1 score metrics on all datasets, which indicates the effectiveness of our framework on document-level RE tasks.

**Document-Level Knowledge Graph Retrieval.** To further investigate how the document-level knowledge graph (DocKG) retrieval effects the performance of LLMs on document-level RE tasks, we conduct extra experiments on the development set of RE-DocRED with different $\tau_{eq} \in [0, 1]$, which is a pre-defined threshold that for relevant subgraphs retrieval. As illustrated in Table 3, $\tau_{eq}$

*Table 2.* Ablation study results on both development (Dev) and test (Test) results with Ign-F1 and F1 scores. The performance declines as the removal of different components.

| Model | DocRED | | | | Re-DocRED | | | |
|---|---|---|---|---|---|---|---|---|
| | Dev | | Test | | Dev | | Test | |
| | Ign-F1 | F1 | Ign-F1 | F1 | Ign-F1 | F1 | Ign-F1 | F1 |
| **DocKS-RAG** | **62.76** | **63.87** | **61.83** | **63.59** | **74.64** | **75.31** | **74.32** | **75.49** |
| w/o DocKG | 58.21 | 58.89 | 57.26 | 58.08 | 68.17 | 68.59 | 68.81 | 69.03 |
| w/o SetRAG | 55.09 | 53.51 | 54.32 | 54.96 | 61.64 | 62.37 | 62.29 | 62.63 |
| w/o Hybrid-Prompt Tuning | 20.46 | 21.07 | 20.53 | 20.94 | 26.15 | 26.87 | 25.89 | 26.32 |

*Table 3.* Performance comparison with different threshold $\tau_{eq}$ in DocKG on development (Dev) set of Re-DocRED. Experimental results achieve peak performance at $\tau_{eq} = 0.7$.

| Ours-DocKG | RE-DocRED (Dev) | | | | | |
|---|---|---|---|---|---|---|
| $\tau_{eq}$ | 0 | 0.2 | 0.5 | **0.7** | 0.9 | 1 |
| F1 | 36.42 | 53.26 | 67.45 | **75.31** | 73.89 | 68.59 |
| Ign-F1 | 32.06 | 49.51 | 63.37 | **74.64** | 73.16 | 68.17 |

*Table 4.* Performance comparison with different threshold $k$ in SetRAG on development (Dev) set of Re-DocRED. Experimental results achieve peak performance at $k = 5$.

| Ours-SetRAG | RE-DocRED (Dev) | | | | | |
|---|---|---|---|---|---|---|
| $k$ | 0 | 1 | 3 | 5 | 8 | 10 |
| F1 | 67.78 | 70.15 | 74.26 | **75.31** | 71.23 | 68.26 |
| Ign-F1 | 65.32 | 68.54 | 72.75 | **74.64** | 71.52 | 68.45 |

has critical influence on the performance of DocKG-RAG. Specifically, at $\tau_{eq} = 0$, DocKS-RAG exhibits the worst performance, indicating that all entities in DocKG are considered relevant to those within the user queries. Such case equals to randomly selecting an entity from DocKG and constructing its corresponding subgraph, which will bring massive noise and lead to poor prediction. As $\tau_{eq}$ increases, both F1 and Ign-F1 scores show substantial improvement and achieve peak performance at $\tau_{eq} = 0.7$, reflecting the effectiveness of graph retrieval by capturing the rich interactions between entities and relations. However, the performance declines when the threshold raises more, which demonstrates that only considering the most relevant entitiy in DocKG limits the global understanding of model.

**Sentence-Level Knowledge Base Retrieval.** We further investigate how SetRAG mechanism effects the performance of LLMs on document-level RE tasks. Extra experiments are performed on the development set of RE-DocRED with different $k \in \mathbb{N}$, which is a pre-defined threshold that means the *Topk* relevant sentences will be retrieved. As demonstrated in Table 4, the parameter $k$ has sufficient influence in the performance of DocKS-RAG. As the threshold increases, both F1 and Ign-F1 scores exhibit significant improvements, reaching peak performance at $k = 5$, which shows that retrieving relevant contextual sentences is helpful to improve the relational and semantic understanding of LLMs for document-level RE tasks. However, as the threshold increases more, lowly relevant or even irrelevant sentences are probably retrieved, which will bring extra noises and lead

to the decline of overall performance and model reliability.

## 5.6. Ablation Study

To assess the effectiveness of each component in our DocKS-RAG framework, we further perform an ablation study by removing different components step by step. As illustrated in Table 2, DocKS-RAG, with all components, achieves the highest performance, showcasing the positive contribution of each component in our proposed framework. As the document-level knowledge graph (DocKG) component is removed, the model's performance declines, which suggests that the integration of structural knowledge is crucial for LLMs to capture the structural information between entities and relations. Then, the removal of the sentence-level semantic Retrieval-Augmented Generation (SetRAG) component results in further performance degradation, which demonstrates that our designed SetRAG mechanism has sufficient influences in enhancing the semantic understanding of LLMs for better extraction. Moreover, such results demonstrate the necessity of our proposed hybrid prompts generation method, which effectively alleviates the misalignment between structural knowledge and semantic information. Furthermore, the removal of Hybrid-Prompt Tuning leads to a drastic reduction in performance than others. In our framework, such case equals to Few-shot learning on LLMs, which means the model only relies on transferred knowledge from few instances and leverages such experience for new tasks. The results shows that few-shot learning method is not suitable for complex document-level RE tasks in our framework, and underscores the necessity of our proposed hybrid-prompt tuning in optimizing the model's expressiveness and accuracy.

## 6. Conclusion and Future Work

In this paper, we propose a novel and effective framework, named DocKS-RAG, which enhances the adaptability and performance of LLMs for complex document-level RE tasks. Extensive experiments on two benchmark datasets demonstrate that our proposed DocKS-RAG significantly outperforms state-of-the-art methods. In the future, we will explore the following problems on how to apply DocKS-RAG to different domains, and how to enhance the scalability of DocKS-RAG with different scales of LLMs.

# 7. Response to Reviewers

## 7.1. Response to reviewer 1:

Q1: Could the authors elaborate on the trade-offs between complexity and efficiency in practical deployments?

Response to Q1: In practical applications, we take full advantage of the parallel processing capabilities provided by both DocKG and SetRAG, enabling us to handle tasks efficiently while maintaining high performance of relational extraction. Additionally, by applying threshold retrieval strategies, we could further simplify the complexity of our operations.

Q2: How DocKS-RAG specifically mitigates the noise during knowledge graph construction and retrieval processes?

Response to Q2: As presented in Section 4, firstly, DocKS-RAG applies a threshold parameter $\tau_{er}$ during graph retrieval to filter out extraneous entities. Secondly, another threshold parameter $\tau_{eq}$ is used to retrieve the relevant contextual sentences. Finally, DocKS-RAG generates hybrid prompts that combine the information from both the DocKG and SetRAG, which integrates relevant contextual information to enhance semantic alignment, improving the overall extraction performance.

## 7.2. Response to reviewer 2:

C1: It might be useful to briefly explain how SetRAG differs from typical retrieval mechanisms.

Response to C1: Our proposed SetRAG module constructs a knowledge base from segmented sentences and retrieves contextually relevant information based on embeddings, allowing for improved understanding of the document's semantics. Such semantic information will be integrated with structural knowledge from DocKG, which further enhances the performance of LLMs in relation extraction tasks.

Q1: How do the authors explain that simpler configurations that do not utilize DocKG or SetRAG yield reasonably good scores?

Response to Q1: As shown in Table 4, the experimental results indicate that the integration of the DocKG and the SetRAG mechanisms significantly enhances model performance, demonstrating the effectiveness of our proposed DocKS-RAG module. In practical deployment scenarios, we capitalize on the parallelization capabilities offered by both DocKG and SetRAG, which allows for efficient processing without compromising the quality of relational extraction. Furthermore, by implementing strategies such as threshold retrieval, we manage to streamline the complexity of the operations involved. Integrating both modules enhances the model's adaptability to various document types and ensures robust performance in real-world applications.

## 7.3. Response to reviewer 3:

Q1: The authors need to explain why they believe graph-based methods lack contextual information and what specific types of context are missing.

Response to Q1: Graph-based methods primarily focus on the structural relationships between entities, which can obscure essential semantic nuances due to the static nature of graph representations. In complex document-level relation extraction tasks, entities may relate across texts, paragraphs, or even chapters within a document. This limitation in the graph-based approach restricts our understanding of how entities interact in different contexts, ultimately hindering the effectiveness of extraction.

Q2: In Eq.(10), the authors introduce a generation function $g_{DocKG}$. How do they implement this function?

Response to Q2: $g_{DocKG}$ is implemented as a function for graph knowledge transformation. We pre-defined various entity-relation mapping types. For example, if a subgraph has "Wisconsin" and "U.S." connected by the edge "state", it would be transformed into the statement "Wisconsin is a state of "U.S." based on our mappings. Ultimately, we concatenate such statements with the relevant semantic information retrieved from SetRAG, and obtain the hybrid prompts.

## 7.4. Response to reviewer 4:

Q1: Could you provide a detailed explanation of the operations of Formula 10 and Formula 15?

Response to Q1: As illustrated in Section 4, the purpose of Formula 10 is to generate informative prompts based on the retrieved entities and relations from DocKG. It defines the prompt generation function $g_{DocKG}$, which takes a set of relevant entities and relations as input and outputs structured prompts $P_{DocKG}$. Formula 15 constructs hybrid prompts using the retrieved sentences from SetKB. It employs the generation function $g_{SetRAG}$ to create prompts from the relevant triplets $T_k$ while appending the retrieved sentences $S_k$. This integration enriches the prompts by combining contextual insights with structured knowledge, enhancing the model's ability to perform document-level relation extraction accurately.

Q2: Can you elaborate on how your framework specifically address the semantic misalignment between PLMs and knowledge graphs?

Response to Q2: As mentioned in Section 4.2, the embedding spaces of PLM and DocKG differ significantly. To address this issue, we integrate PLM embeddings as the initial inputs for entities and relations, which allows DocKG to learn from the PLM during the updating process and improves the reliability of similarity calculations.

## Acknowledgments

This work was supported in part by the Jiangsu Provincial Major Project on Basic Research of Cutting-edge and Leading Technologies, under grant no. BK20232032 and National Natural Science Foundation of China under grant no. 92267104.

## Impact Statement

This paper presents work whose goal is to advance the field of Deep Learning. There are many potential societal consequences of our work, none which we feel must be specifically highlighted here.

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

# A. Appendix

## A.1. Datasets

We evaluate our proposed DocKS-RAG on two widely used benchmark datasets in document-level RE, which are shown as follows:

- **DocRED** (Yao et al., 2019) is a common benchmark for the document-level RE tasks, comprising a diverse set of documents extracted from Wikipedia. The goal of this dataset is to extract the structured triplets from documents and it allows for the evaluation of models' abilities to identify relations by cross-sentence, cross-paragraph and even cross-discourse, which reflects the complexities inherent in real-world documents.

- **Re-DocRED** (Tan et al., 2022a) is an advanced extension of the original DocRED dataset. It retains the rich and diverse annotations of entity pairs and their relations from its predecessor DocRED, and introduces a relational reasoning aspect that enables models to better capture the complex interactions and dependencies between entities and relations from extensive documentation data.

## A.2. Baselines

We evaluate our proposed framework by comparing with recently presented PLMs-based, graph-based and LLMs-based methods, which are listed as follows:

- HIN (Tang et al., 2020a) tackles the challenge in multigranularity inference by leveraging different levels of contextual information for accurate extraction.

- ATLOP-BERT (Zhou et al., 2020) obtains a more comprehensive representation of the entities by transferring attention from the PLMs to capturing the associated contexts.

- SRLR (Huang et al., 2023) tackles the challenges in the relation representation and logical reasoning by extending the extraction to the mention level.

- GLRE (Wang et al., 2020) proposes a method to encode document information through global and local representation of entities and aggregate with the context information for better extraction.

- LSR (Nan et al., 2020) particularly considers capturing the complex interactions between inter-sentence entities and employing a refinement strategy for multi-hop reasoning.

- HeterGSAN (Xu et al., 2020) proposes the reconstruction of ground-truth path dependencies and focuses on the relevant entity pairs to improve the accuracy of RE.

- SIRE (Zeng et al., 2021) mitigates the negative influence caused by indiscriminate representation by differentiating the representation of relations and implementing a comprehensive logical reasoning module.

- SSAN (Xu et al., 2021a) proposes two alternative transformation modules to produce attentive biases for adaptively regularization.

- DocuNet (Zhang et al., 2021) introduces an entity-level relation matrix and incorporates both context information and global dependencies through a U-shaped segmentation module.

- KD-DocRE (Tan et al., 2022b) adopts a knowledge distillation method to deal with the discrimination between the annotated texts and distantly supervised data.

- KMGRE (Jiang et al., 2022) presents a mention-level relation extractor to solve the multi-label problem in optimizing the mention-level relation extractor.

- AutoRE (Xue et al., 2024) presents an innovative relational extraction paradigm for LLMs that obviates the dependence on pre-defined relations and emphasizes the identification of triplet facts.

- LMRC (Li et al., 2024) addresses the performance gap in between LLMs and traditional approaches by proposing a novel classifier-LLM methodology that effectively directs the LLM's attention towards relevant entity pairs.

## A.3. Implementation Details

The technical modules of DocKS-RAG primarily consist of document-Level knowledge graph construction, sentence-level knowledge base construction, and hybrid-prompt tuning. The specific experimental parameters are set as follows:

- **Document-Level Knowledge Graph Construction and Retrieval.** We employ general graph neural network for the construction of Document-level Knowledge graph (DocKG) and later extract structural knowledge from DocKG by graph retrieval for further hybrid prompt generation. In the stage of construction, both the entity and relation embedding dimensions are 128, and the network is initialized with 8 convolutional layers, which aggregate the neighborhood information and capture the long-range dependencies within the document. Dropout is set at a rate of 0.1 in the convolutional layers to mitigate the risk of overfitting. To enhance the stability of training, *Adam* is employed as the optimizer. Learning rate is set to 1e-3, and weight decay is 5e-4. In the stage of retrieval, we establish a minimum threshold at 0.7 for measuring the similarity,

which guarantees that only highly relevant subgraphs will be extracted. We transfer the relevant subgraphs to informative prompts with a maximum length of 128 tokens.

- **Sentence-Level Knowledge Base Construction and Semantic Retrieval.** We harness BGE as embedding model for sentence representation. In the semantic retrieval component, *Top5* relevant sentences will be retrieved associated with the user query, which are further transferred into informative prompts with maximum length of 512 tokens.

- **Hybrid-Prompt Tuning.** We adopt LLaMA3-8B as the backbone LLM. $LORA$ is chosen as the adapter training method to fine-tune LLaMA3-8B. The low-rank setting of the adapter is 16, along with the warmup ratio of 0.1. We set the learning rate to 5e-5, and the batch size is 2 for both the training and evaluation stage. We utilize a learning rate scheduler with cosine variations for training, and we conduct fine-tuning for 3 epochs. All experiments conducted in this paper are implemented by PyTorch, and are trained on four 24GB RTX 3090 GPUs.

