# OpenReview forum: "DocKS-RAG: Optimizing Document-Level Relation Extraction through LLM-Enhanced Hybrid Prompt Tuning"
_ICML.cc/2025/Conference — ICML 2025 poster_

### Official Review · Reviewer_RKaC · 2025-03-11

**Overall Recommendation:** 4

**Summary:**

The authors of the paper propose a novel approach for document-level relation extraction. During the training phase, they prepare two additional texts: one sourced from DocKG and the other from SetRAG, which are concatenated and utilized as a prefix in the final prompt. Subsequently, they fine-tune a small open-source language model to predict the relations present from the input document.

**Claims And Evidence:**

Yes. The experiments conducted are on specific benchmark datasets; hence, the generalizability of the results to other domains or types of documents remains somewhat unspecified.

**Essential References Not Discussed:**

NA.

**Experimental Designs Or Analyses:**

Yes, the experiment design and analysis is suitable.

**Methods And Evaluation Criteria:**

Yes. The selection of benchmark datasets specific to document-level relation extraction is appropriate. Both datasets are well-established in the field and are designed to evaluate models on complex, multi-entity document scenarios.

**Other Comments Or Suggestions:**

1)For further understandability, it would also makes sense to put some numbers in Figure 2 for each step e.g. GNN training which are then repeated in the corresponding section to guide the reader a bit more.
2)Although the paper provides a detailed methodology, some sections could benefit from greater clarity. For instance, the hybrid prompt generation process may require more concrete examples to fully illustrate the differences and advantages over conventional methods.
3)Could you provide a detailed explanation of the operations of Formula 10 and Formula 15?

**Other Strengths And Weaknesses:**

Strengths:
1)The paper is well-organized, with a clear delineation of the problem, proposed methodology, and experimental results.
2)The authors present their ideas logically, making it accessible for readers who may not have extensive background knowledge in document-level relation extraction.

Weaknesses:
1)The hybrid prompt generation process is complex and may be difficult for users unfamiliar with the underlying methodologies to interpret or adjust, which could limit its accessibility.
2)The emphasis on document-level relation extraction may not fully address the challenges posed by unstructured, noisy, or overly complex input data.

**Questions For Authors:**

1)In your results, you mention challenges related to semantic misalignment between PLMs and knowledge graphs. Can you elaborate on how your framework specifically addresses this issue?
2)The authors propose a hybrid prompts generation method that combines structural interactions with semantic information. Could you clarify the process for generating these hybrid prompts, and perhaps provide examples of how they differ from standard prompts?

**Relation To Broader Scientific Literature:**

Unlike traditional methods that often rely heavily on fine-tuning PLMs, the paper innovatively proposes a hybrid approach that enhances model adaptability by combining structural knowledge with contextual prompts, thus reflecting a significant evolution from earlier works that typically did not fully harness the potential of LLMs in this context.

**Theoretical Claims:**

Yes. The choice of using F1 and Ign-F1 scores as evaluation metrics is justified within the context of document-level RE tasks. The theoretical underpinning of why these metrics are particularly suitable for assessing the model's capabilities in capturing implicit relations could be discussed more rigorously.

---

> ### Author Rebuttal · Authors · 2025-03-31
>
> Thank you for your valuable feedback.
>
> C1: Although the paper provides a detailed methodology, some sections could benefit from greater clarity. For instance, the hybrid prompt generation process may require more concrete examples to fully illustrate the differences and advantages over conventional methods.
>
> Response to C1: We appreciate your feedback, particularly regarding the clarity of the hybrid prompt generation process. We aim to enrich this section by including specific examples that illustrate how the generated prompts differ from those produced by conventional methods. By showcasing practical scenarios, we intend to clarify the advantages of our hybrid approach and its effectiveness in providing richer context for relation extraction tasks.
>
> Q1: Could you provide a detailed explanation of the operations of Formula 10 and Formula 15?
>
> Response to Q1: Thank you for your question. As illustrated in Section 4, the purpose of Formula 10 is to generate informative prompts based on the retrieved entities and relations from DocKG. It defines the prompt generation function $ g_{DocKG} $, which takes a set of relevant entities and relations as input and outputs structured prompts $ P_{DocKG} $​.  Formula 15 constructs hybrid prompts using the retrieved sentences from SetKB. It employs the generation function $ g_{SetRAG} $​ to create prompts from the relevant triplets $ T_k $​ while appending the retrieved sentences $ S_k $​. This integration enriches the prompts by combining contextual insights with structured knowledge, enhancing the model's ability to perform document-level relation extraction accurately.
>
> Q2: In your results, you mention challenges related to semantic misalignment between PLMs and knowledge graphs. Can you elaborate on how your framework specifically addresses this issue?
>
> Response to Q2: Thank you for your thoughtful insights. As mentioned in Section 4.2, the embedding spaces of PLM and DocKG differ significantly. To address this issue, we integrate PLM embeddings as the initial inputs for entities and relations, which allows DocKG to learn from the PLM during the updating process and improves the reliability of similarity calculations.
>
> Q3: Could you clarify the process for generating these hybrid prompts, and perhaps provide examples of how they differ from standard prompts?
>
> Response to Q3: Thank you for your insightful feedback. As shown in Section 4.4, we incorporate both relevant structural and semantic information from DocKG and SetRAG. Specifically, we pre-defined various entity-relation mapping types. For example, if a subgraph has "Wisconsin" and "U.S." connected by the edge "state", it would be transformed into the statement "Wisconsin is a state of U.S." based on our mappings. Ultimately, we concatenate such statements with the relevant semantic information retrieved from SetRAG, and obtain the hybrid prompts.

---

### Official Review · Reviewer_V77s · 2025-03-11

**Overall Recommendation:** 3

**Summary:**

In this paper, the authors propose a DocKS-RAG method to combine structural knowledge and semantic information for document-level relation extraction task. In DocKS-RAG, the authors first rely on GNNs to construct a document-Level knowledge graph and retrieve relevant information from this graph according to the user query. Then, they extract relevant sentences from the input document. On this basis, they construct hybrid prompts and adopt LORA to train an LLM. Extensive experiments on two datasets verify the effectiveness of the proposed method.

**Claims And Evidence:**

The experiments on two widely-used benchmarks demonstrate the effectiveness of the method. Additionally, the ablation study shows the importance of each component in the method.

**Essential References Not Discussed:**

No

**Experimental Designs Or Analyses:**

Yes, I have checked the details in Section 5, including the experimental setup and result analyses. The experiments in Section 5.5 verify the overall effectiveness and the influence of each hyper-parameter, while the results in Section 5.6 verify the importance of each module in the proposed method.

**Methods And Evaluation Criteria:**

I believe that the proposed method is suitable for the problem. The motivation for combing structural knowledge and semantic information makes sense and is very important. As for knowledge construction, the proposed GNN-based method adopts a general pipeline in existing literature. As for semantic information, the extraction process is also acceptable. Additionally, the benchmarks and metrics are reasonable.

**Other Comments Or Suggestions:**

There are some typos in this paper, such as:

1.	Line 120, “model” should be “models”

2.	Line 351, “a extra” should be “an extra”

**Other Strengths And Weaknesses:**

The main strengths of this paper include:

1.	The motivation of combining structural knowledge and semantic information makes sense to me and is important.

2.	The proposed method can effectively achieve the goal in this paper.

3.	The experimental comparison with 14 baselines (including some SOTA methods) is solid and convincing.

The main weakness of this paper include:

1.	Many technical details need to be further illustrated (please see the questions 3-5 below).

2.	The technical contributions of this paper seem to be limited. The methods used to extract knowledge and relevant sentences in the document are very common in the literature. I don’t find essential improvements of the method.

3.	The experimental section needs to be further revised (please see the question 6 below).

**Questions For Authors:**

1.	In Introduction, the authors state that “graph-based methods … lack of sufficient contextual information”. The authors need to explain why they believe graph-based methods lack contextual information and what specific types of context are missing.

2.	Also in Introduction, the authors state that “PLMs are … the semantic misalignment between the texts and graphs”. I don’t understand how they address this issue in this paper.

3.	In Section 4.2, the proposed method needs to train entity pair representations to construct the document-level knowledge graph. Where do the training samples come from? Besides, will this process limit the applicability of the method to other domains?

4.	In Eq.(10), the authors introduce a generation function g_{DocKG}. How do they implement this function?

5.	When explaining the extraction of knowledge graphs from the knowledge base, the authors mention the user query q. However, in Section 3 (Problem Definition), there is no indication that a user query is provided, and in Figure 1, the query is also not visible. The authors need to clarify this inconsistency.

6.	In Section 5, the authors provide the hyper-parameter experiments before the ablation study, which seems to be weird for me. I suggest rearranging the order to improve clarity and coherence

**Relation To Broader Scientific Literature:**

In my opinion, the proposed method provides a general perspective to address the issues of document-level RE task. The idea of integrating structural knowledge and semantical information is also general enough to extend to our domains or tasks.

**Theoretical Claims:**

This paper does not make any theoretical claims.

---

> ### Author Rebuttal · Authors · 2025-03-31
>
> We sincerely thank the reviewer for recognizing and affirming our work. Regarding the questions raised during the review process, we have carefully considered them and provided detailed responses as follows:
>
> Q1: In Introduction, the authors state that “graph-based methods … lack of sufficient contextual information”. The authors need to explain why they believe graph-based methods lack contextual information and what specific types of context are missing.
>
> Response to Q1: Thank you for your valuable question. Graph-based methods primarily focus on the structural relationships between entities, which can obscure essential semantic nuances due to the static nature of graph representations. In complex document-level relation extraction tasks, entities may relate across texts, paragraphs, or even chapters within a document. This limitation in the graph-based approach restricts our understanding of how entities interact in different contexts, ultimately hindering the effectiveness of extraction.
>
> Q2: Also in Introduction, the authors state that “PLMs are … the semantic misalignment between the texts and graphs”. I don’t understand how they address this issue in this paper.
>
> Response to Q2: As shown in Section 4.4, we integrate informative structural knowledge from DocKG with the relevant sentences retrieved by SetRAG, and further generate the hybrid prompts to enhance the interaction between the textual data and the graph-based representations, thereby improving relation extraction performance.
>
> Q3: Where do the training samples come from? Besides, will this process limit the applicability of the method to other domains?
>
> Response to Q3: Thank you for your insightful questions. The training samples for constructing DocKG come from observable documentation data, which is utilized to obtain representations of entities and relations. In our proposed DocKS-RAG framework, the training process entails the construction of DocKG and SetKB. Relevant subgraphs and sentences are retrieved to create hybrid prompts, which are then fine-tuned using PEFT to improve the performance of LLMs in document-level relation extraction tasks. Although DocKS-RAG relies on this training process, it is adaptable and can be applied to other domains by fine-tuning the model with domain-specific data.
>
> Q4: In Eq.(10), the authors introduce a generation function $ g_{DocKG} $. How do they implement this function?
>
> Response to Q4: Thank you for your thoughtful insights. $ g_{DocKG} $ is implemented as a function for graph knowledge transformation. We pre-defined various entity-relation mapping types. For example, if a subgraph has "Wisconsin" and "U.S." connected by the edge "state", it would be transformed into the statement "Wisconsin is a state of U.S." based on our mappings. Ultimately, we concatenate such statements with the relevant semantic information retrieved from SetRAG, and obtain the hybrid prompts.
>
> C1: When explaining the extraction of knowledge graphs from the knowledge base, the authors mention the user query q. However, in Section 3 (Problem Definition), there is no indication that a user query is provided, and in Figure 1, the query is also not visible. The authors need to clarify this inconsistency.
>
> Response to C1: As mentioned in  Section 4, q is denoted as the input user query for further extraction. We appreciate your insight and will consider including the explanations about q in the Problem Definition section and Figure 1 in the revised manuscript to enhance the readability.
>
> C2: In Section 5, the authors provide the hyper-parameter experiments before the ablation study, which seems to be weird for me. I suggest rearranging the order to improve clarity and coherence.
>
> Response to C2: Thank you for your valuable suggestion. We will consider making this change to the revised version.

---

### Official Review · Reviewer_Amvq · 2025-03-13

**Overall Recommendation:** 4

**Summary:**

In this work, the authors introduce DocKS-RAG, a framework that enhances large language models for document-level relation extraction. By integrating structural knowledge from a Document-level Knowledge Graph (DocKG) with semantic insights from a Sentence-level Semantic Retrieval-Augmented Generation (SetRAG) mechanism, the framework effectively captures complex relationships in documents. The paper emphasizes the importance of aligning structural and semantic knowledge to address the noise associated with traditional methods. Experiments on DocRED and Re-DocRED demonstrate that DocKS-RAG significantly improves accuracy and highlights the advantages of hybrid-prompt tuning techniques.

**Claims And Evidence:**

The authors claim that their proposed framework, DocKS-RAG, addresses limitations in existing document-level relation extraction approaches by effectively combining linguistic and structural knowledge. Extensive experiments demonstrate that DocKS-RAG achieves superior performance metrics compared to state-of-the-art methods, as evidenced by significant gains in F1 and Ign-F1 scores. Additionally, the systematic ablation studies offered in the paper highlight the significance of each component, fostering confidence in the robustness of the framework's design and functionality.

**Essential References Not Discussed:**

The references are enough.

**Experimental Designs Or Analyses:**

An ablation study is conducted by removing different components of the DocKS-RAG framework, showing the impact of each component on performance. However, the nuances of the results could be better articulated—explaining why the removal of specific components leads to performance degradation would provide deeper insights into the framework's mechanics.

**Methods And Evaluation Criteria:**

In this paper, the authors use a Document-level Knowledge Graph (DocKG) alongside a Sentence-level Semantic Retrieval-Augmented Generation (SetRAG) mechanism, which allows the framework to capture both the structural relationships among entities and the contextual semantics of the text. This dual approach addresses the semantic misalignment typically observed between pre-trained language models and knowledge graphs, which is particularly crucial for complex document-level tasks.

**Other Comments Or Suggestions:**

When mentioning the sentence-level Semantic Retrieval-Augmented Generation, it might be useful to briefly explain how it differs from typical retrieval mechanisms.

**Other Strengths And Weaknesses:**

The idea of combining multiple retrieval and KG creation methods to form a RAG-based approach with heterogeneous data is interesting. The main novelty stems from the DocKG approach. The DocKG is generated by predicting if a relation should be added between two entities, given the embedding of the entities and the relation. The authors should ensure that all acronyms such as PLMs and KGs have been defined the first time they appear. This aids readers who may be unfamiliar with these terms.

**Questions For Authors:**

Although DocKS-RAG demonstrates high performance, the ablation studies indicate that competitive results can still be achieved without integrating structural and semantic components. For instance, simpler configurations that do not utilize DocKG or SetRAG yield reasonably good scores. Could the authors elaborate on the trade-offs between complexity and efficiency in practical deployments?

**Relation To Broader Scientific Literature:**

The study's emphasis on integrating structural knowledge via Document-level Knowledge Graphs (DocKG) echoes findings in the literature that suggest the synergy between PLMs and knowledge graphs (KGs) can improve contextual understanding. The approach taken in this paper thus not only aligns with but also advances previous investigations into the necessity of structural information in processing complex relationships within texts.

**Theoretical Claims:**

The paper asserts that employing a hybrid-prompt tuning approach, coupled with Parameter-Efficient Fine-Tuning (PEFT), leads to improved adaptability and performance of LLMs. This claim remains largely substantiated by experimentation.

---

> ### Author Rebuttal · Authors · 2025-03-31
>
> Thank you for your positive feedback on our paper.
>
> C1: When mentioning the sentence-level Semantic Retrieval-Augmented Generation, it might be useful to briefly explain how it differs from typical retrieval mechanisms.
>
> Response to C1: Thank you for your valuable comment. Our proposed SetRAG module constructs a knowledge base from segmented sentences and retrieves contextually relevant information based on embeddings, allowing for improved understanding of the document's semantics. Such semantic information will be integrated with structural knowledge from DocKG, which further enhances the performance of LLMs in relation extraction tasks.
>
> Q1: Although DocKS-RAG demonstrates high performance, the ablation studies indicate that competitive results can still be achieved without integrating structural and semantic components. For instance, simpler configurations that do not utilize DocKG or SetRAG yield reasonably good scores. Could the authors elaborate on the trade-offs between complexity and efficiency in practical deployments?
>
> Response to Q1: Thank you for your thoughtful question. As shown in Table 4, the experimental results indicate that the integration of the DocKG and the SetRAG mechanisms significantly enhances model performance, demonstrating the effectiveness of our proposed DocKS-RAG module. In practical deployment scenarios, we capitalize on the parallelization capabilities offered by both DocKG and SetRAG, which allows for efficient processing without compromising the quality of relational extraction. Furthermore, by implementing strategies such as threshold retrieval, we manage to streamline the complexity of the operations involved. Integrating both modules enhances the model's adaptability to various document types and ensures robust performance in real-world applications.

---

### Official Review · Reviewer_CMNB · 2025-03-13

**Overall Recommendation:** 4

**Summary:**

The paper introduces DocKS-RAG, a novel framework aimed at enhancing document-level relation extraction (RE) by integrating large language models (LLMs) with structured knowledge graphs. The proposed method combines a Document-level Knowledge Graph (DocKG) with a Sentence-level Semantic Retrieval-Augmented Generation (SetRAG) mechanism to improve entity-relation understanding in complex documents. The authors conduct extensive experiments on benchmark datasets, DocRED and Re-DocRED, demonstrating that DocKS-RAG significantly outperforms existing PLM-based and graph-based methods by achieving superior F1 and Ign-F1 scores, thereby validating its effectiveness in addressing the intricacies of document-level RE tasks.

**Claims And Evidence:**

The claims made in the submission regarding the efficacy and advantages of the DocKS-RAG framework in document-level relation extraction (RE) appear to be generally supported by clear and convincing evidence. The authors substantiate their claims through extensive experimental results, comparing their framework against existing state-of-the-art methods on benchmark datasets like DocRED and Re-DocRED.

**Essential References Not Discussed:**

None

**Experimental Designs Or Analyses:**

The authors compare DocKS-RAG against several established methods, including both PLMs-based and graph-based approaches. This comparative analysis is essential for establishing the framework's effectiveness.

**Methods And Evaluation Criteria:**

Conducting ablation studies to evaluate the contribution of different components validates the efficacy of each aspect of the framework. This methodological rigor strengthens the argument for the importance of integrating both structural and contextual information in improving extraction performance.

**Other Comments Or Suggestions:**

Since the paper targets a broader audience, adding a small section that explains potential practical applications and user scenarios for the proposed framework could make the contributions more relatable. People appreciate understanding how theoretical advancements may impact real-world applications.

**Other Strengths And Weaknesses:**

Strengths:
- The paper addresses a critical gap in natural language processing by focusing on document-level relation extraction, which is vital for tasks like knowledge graph construction and information retrieval.
- Extensive experiments on DocRED and Re-DocRED benchmarks are robust, showcasing superior performance in both F1 and Ign-F1 scores compared to state-of-the-art methods.
- Comprehensive ablation studies clarify the contributions of individual components (DocKG, SetRAG, and hybrid-prompt tuning), reinforcing the validity of the design choices.

Weakness:
- While the authors highlight the novelty of combining graph-based and LLM-based methods, the range of comparison baselines is limited. Existing works on the BioRED [1] dataset and similar benchmarks in knowledge extraction provide a richer set of methods that could offer additional insights into the framework’s relative performance. Incorporating these comparisons could strengthen the claim of novelty and demonstrate broader applicability.
- While DocKS-RAG achieves high performance, the ablation studies suggest that even without blending structural and semantic components, competitive results can be obtained. For example, simpler setups without DocKG or SetRAG achieve reasonably good scores. Could the authors elaborate on the trade-offs between complexity and efficiency in practical deployments?

Reference:
[1] Islamaj, Rezarta, et al. "The overview of the BioRED (Biomedical Relation Extraction Dataset) track at BioCreative VIII." Database 2024 (2024): baae069.

**Questions For Authors:**

- In your discussion, you mentioned that existing graph-based methods can introduce noise and irrelevant connections, affecting performance. Can you elaborate on how DocKS-RAG specifically mitigates this noise during knowledge graph construction and retrieval processes?
- Regarding future work, how do you envision scaling DocKS-RAG to accommodate different types of documents or domains with vastly different structure or language use (e.g., legal texts, scientific literature)? What modifications might be necessary to adapt the framework effectively?

**Relation To Broader Scientific Literature:**

Prior to this study, substantial efforts were made to enhance relation extraction primarily through sentence-level methods using pre-trained language models (PLMs), such as BERT (Devlin et al., 2019) and REBEL (Huguet Cabot & Navigli, 2021). The present paper's introduction of a document-level approach, specifically through the DocKS-RAG framework, builds on this prior work by addressing the limitations of existing PLMs-based methods, thus contributing to ongoing dialogues about enhancing RE capabilities.

**Theoretical Claims:**

This paper primarily focuses on empirical methodologies rather than formal theoretical proofs. However, it does include theoretical claims regarding the effectiveness of the proposed methods, particularly the integration of structural knowledge and contextual understanding through the DocKS-RAG framework.

---

> ### Author Rebuttal · Authors · 2025-03-31
>
> Thank you for your positive comments and insightful questions.
>
> Q1: While DocKS-RAG achieves high performance, the ablation studies suggest that even without blending structural and semantic components, competitive results can be obtained. For example, simpler setups without DocKG or SetRAG achieve reasonably good scores. Could the authors elaborate on the trade-offs between complexity and efficiency in practical deployments?
>
> Response to Q1: We appreciate your insightful question. In practical applications, we take full advantage of the parallel processing capabilities provided by both DocKG and SetRAG, enabling us to handle tasks efficiently while maintaining high performance of relational extraction. Additionally, by applying threshold retrieval strategies, we could further simplify the complexity of our operations.
>
> Q2: In your discussion, you mentioned that existing graph-based methods can introduce noise and irrelevant connections, affecting performance. Can you elaborate on how DocKS-RAG specifically mitigates this noise during knowledge graph construction and retrieval processes?
>
> Response to Q2: Thank you for your valuable feedback. As presented in Section 4, firstly, DocKS-RAG applies a threshold parameter $ \tau_{er} $ during graph retrieval to filter out extraneous entities. Secondly, another threshold parameter $ \tau_{eq} $ is used to retrieve the relevant contextual sentences. Finally, DocKS-RAG generates hybrid prompts that combine the information from both the DocKG and SetRAG, which integrates relevant contextual information to enhance semantic alignment, improving the overall extraction performance.
>
> Q3: Regarding future work, how do you envision scaling DocKS-RAG to accommodate different types of documents or domains with vastly different structure or language use (e.g., legal texts, scientific literature)? What modifications might be necessary to adapt the framework effectively?
>
> Response to Q3: To scale DocKS-RAG for diverse document types or domains, modifications could include tailoring the DocKG construction to account for domain-specific terminology and relationships, alongside enhancing the SetRAG mechanism to accommodate varied syntactical structures and language use. Additionally, training additional domain-specific models or fine-tuning existing ones with targeted datasets would improve adaptability and semantic accuracy for specific contexts.

---

### Decision · Program_Chairs · 2025-05-01

**Decision:**

Accept (poster)

**Comment:**

This paper presents DocKS-RAG, a novel framework that enhances document-level relation extraction through the integration of structural knowledge from a Document-level Knowledge Graph (DocKG) with semantic information from a Sentence-level Semantic Retrieval-Augmented Generation (SetRAG) mechanism. It effectively addresses limitations in current approaches by combining linguistic and structural knowledge in an innovative way.

I recommend accepting this paper as three reviewers all gave positive comments. The integration of structural knowledge with contextual understanding is both innovative and well-justified. The experimental results are consistently good across established benchmarks, with comprehensive ablation studies validating the contribution of each component.